# Viscoelastic Hemostatic Assays for Orthopedic Trauma and Elective Procedures

**DOI:** 10.3390/jcm11144029

**Published:** 2022-07-12

**Authors:** Christiaan N. Mamczak, Jacob Speybroeck, John E. Stillson, Joseph Dynako, Andres Piscoya, Ethan E. Peck, Michael Aboukhaled, Emily Cancel, Michael McDonald, Diego Garcia, John Lovejoy, Stephanie Lubin, Robert Stanton, Matthew E. Kutcher

**Affiliations:** 1Department of Orthopaedic Trauma, Florida Orthopaedic Institute, St. Petersburg, FL 33701, USA; cmamczak@floridaortho.com; 2Department of Orthopaedic Surgery, MetroHealth Medical Center, Case Western Reserve University School of Medicine, Cleveland, OH 44106, USA; jacob.speybroeck@uhhospitals.org; 3Department of Family Medicine, John Peter Smith Hospital, Fort Worth, TX 76104, USA; 4Department of Orthopaedic Surgery, University of Mississippi Medical Center, Jackson, MS 39216, USA; jdynako@umc.edu; 5Department of Orthopaedic Surgery, Walter Reed National Military Medical Center, Bethesda, MD 20814, USA; apiscoya23@gmail.com; 6Department of Emergency Medicine, St. Joseph Regional Medical Center, Mishawaka, IN 46545, USA; ethan.peck@valpo.edu (E.E.P.); maboukhaled2@gmail.com (M.A.); emlcancel@aim.com (E.C.); dgarciadb5@gmail.com (D.G.); 7Department of Graduate Medical Education, Naval Medical Readiness and Training Command, Portsmouth, VA 23708, USA; mcdonald.mj07@gmail.com; 8Department of Orthopaedic Surgery, University of Florida Health Jacksonville, Jacksonville, FL 32209, USA; jfl264@me.com; 9Department of Orthopaedic Surgery, Hôpital Sacré Coeur, Milot 1230, Haiti; stephlubin@yahoo.com; 10Department of Pediatric Orthopaedics, Nemours Children’s Health-Florida, Orlando, FL 32827, USA; robert.stanton@nemours.org; 11Departments of Surgery and Emergency Medicine, University of Mississippi Medical Center, Jackson, MS 39216, USA; mkutcher@umc.edu

**Keywords:** blood coagulation tests, blood transfusion, orthopedics, thromboelastography, rotational thromboelastometry, venous thromboembolism, trauma

## Abstract

The application of viscoelastic hemostatic assays (VHAs) (e.g., thromboelastography (TEG) and rotational thromboelastometry (ROTEM)) in orthopedics is in its relative infancy when compared with other surgical fields. Fortunately, several recent studies describe the emerging use of VHAs to quickly and reliably analyze the real-time coagulation and fibrinolytic status in both orthopedic trauma and elective orthopedic surgery. Trauma-induced coagulopathy—a spectrum of abnormal coagulation phenotypes including clotting factor depletion, inadequate thrombin generation, platelet dysfunction, and dysregulated fibrinolysis—remains a potentially fatal complication in severely injured and/or hemorrhaging patients whose timely diagnosis and management are aided by the use of VHAs. Furthermore, VHAs are an invaluable compliment to common coagulation tests by facilitating the detection of hypercoagulable states commonly associated with orthopedic injury and postoperative status. The use of VHAs to identify hypercoagulability allows for an accurate venous thromboembolism (VTE) risk assessment and monitoring of VTE prophylaxis. Until now, the data have been insufficient to permit an individualized approach with regard to dosing and duration for VTE thromboprophylaxis. By incorporating VHAs into routine practice, orthopedic surgeons will be better equipped to diagnose and treat the complete spectrum of coagulation abnormalities faced by orthopedic patients. This work serves as an educational primer and up-to-date review of the current literature on the use of VHAs in orthopedic surgery.

## 1. Introduction

Orthopedic surgical patients can display a spectrum of coagulopathies, from hypocoagulopathic pelvic fracture trauma patients in shock requiring massive transfusion to postoperative hypercoagulopathic elective arthroplasty patients in need of deep vein thrombosis (DVT) prophylaxis. Until recently, there have been insufficient data to permit an individualized approach with bedside point-of-care precision testing to guide orthopedic surgeons through resuscitation and VTE prophylaxis. However, viscoelastic hemostatic assays (VHAs, including thromboelastography (TEG) and rotational thromboelastometry (ROTEM)) enable the identification and individualized treatment of a dynamic spectrum of coagulation phenotypes. Blood component therapy (BCT) and hemostatic adjunctive therapy (HAT) to address hypocoagulopathy, as well as anticoagulant prophylaxis to address hypercoagulopathy, are both efficiently directed by VHAs utilizing simple algorithms (see Table 1 below).

An estimated, 25% of severely injured patients arrive in emergency departments with trauma-induced coagulopathy (TIC) [1]; this relatively high incidence of acute hypocoagulopathy in trauma patients has driven an expanded investigation into the use of VHAs. TIC has been reported to be an independent predictor of mortality [2], as well as a risk factor for multisystem organ failure and higher transfusion requirements [3,4,5]. Given the point-of-care nature and fast turn-around time for results, many trauma centers consider VHAs as the standard of care for all severely injured trauma patients on arrival to an emergency department and/or trauma center. There is a large and expanding body of literature describing the utilization of VHAs to better characterize the spectrum of coagulopathy at presentation and the evolving changes through the resuscitation of trauma patients [6].

Utilizing VHAs to guide goal-directed resuscitation and transfusion in trauma patients with coagulopathy may improve patient-centered care and outcomes. A 2016 Cochrane review found that VHA-guided transfusion strategies reduced blood product utilization and improved mortality rates [7]. However, the studies reviewed predominantly involved cardiothoracic surgery patients, highlighting the need for more research in the area of orthopedic injury and polytrauma. Of existing trauma trials, several studies have demonstrated decreased BCT administration and noninferior or improved mortality when using VHA to guide resuscitation [8,9,10,11,12,13]. Randomized controlled trials have demonstrated the use of VHAs to guide BCT and HAT not just in trauma, but also in the management of bleeding patients in the critical care setting [13,14,15,16,17,18]. However, the optimal use of VHAs in resuscitation is not entirely settled: for example, the implementing treatment algorithms for the correction of trauma-induced coagulopathy (ITACTIC) study found no difference in mortality between VHAs and common coagulation tests (CCTs) (e.g., partial thromboplastin time (PTT), prothrombin time (PT), international normalized ratio (INR), platelet count, and fibrinogen level) [19]. The trial investigators detected a lower-than-expected incidence of TIC, although they did identify a mortality benefit in the subgroup of patients with traumatic brain injury. In summary, continued trauma research on the utilization of VHA-guided resuscitation protocols must be implemented.

There is relatively little literature concerning VHAs in orthopedic trauma. From the earliest systematic description of the use of VHA in modern trauma, TEG identified both early hypercoagulability, as well as early hypocoagulability, and transfusion requirements [20]. There was a significant gap of nearly 20 years before the value of VHAs in guiding BCT for hemorrhage in orthopedic trauma pelvic fracture patients was presented [21,22,23]. Beyond its well-described utility in the treatment of hypocoagulopathy, the ability of VHAs to identify hypercoagulopathy has drawn recent attention. More than 85% of injured patients become detectably hypercoagulable by VHA after trauma, and this hypercoagulability is associated with a two-fold increase in the risk of VTE during recovery [24]. VTE rates remain as high as 28% in modern series [25], exceeding 15% even under the most stringent prophylaxis regimens [24], with the presence of orthopedic injury being one of the most influential risk factors. Despite increasing attention paid to VHAs in orthopedic trauma, however, there remains only a single orthopedic-specific review paper on this topic [26].

A barrier to the use of VHA, not just in trauma but in orthopedic surgery, has been an excessive reliance on CCTs to help identify bleeding patterns in patients with musculoskeletal injuries, and a lack of familiarity among orthopedic traumatologists on the indications for and interpretation of these tests. Therefore, a straightforward summary of VHA testing is in order.

### 1.1. Thromboelastography (TEG)

To perform a first-generation TEG assay (using the TEG 5000 Hemostasis Analyzer: Haemonetics, Braintree, MA, USA), a 0.36 mL citrated sample of whole blood is placed into a heated cup maintained at 37 °C, and citrate anticoagulation is reversed by the addition of calcium. A coagulation activator (such as kaolin for a standard TEG, or tissue factor-containing reagent for a “rapid” TEG) is added to initiate coagulation; “native” coagulation can also be measured when the assay is activated by anticoagulant reversal alone. The sample cup rotates 4.45 degrees every ten seconds, with a concentric pin linked to a sensor suspended in the cup (Figure 1). In viscous, unclotted blood, the pin does not move; however, as a clot begins to form, elastic clot fibers bind the cup to the pin, generating rotational forces on the pin. These forces are transmitted to an electrical transducer, producing a graphical output that represents clot dynamics over time. The five major parameters of TEG are reaction time (R), clot kinetics (K), alpha angle, maximum amplitude (MA), and lysis at 30 min (LY30) [27,28].

The second-generation TEG 6s Hemostasis Analyzer (manufacturer: Haemonetics, Braintree, MA, USA) is a fully automated, cartridge-based test designed for ease of use and reproducibility. It has been cleared by the FDA for use in the trauma setting based on a recent multicenter method comparison trial which included over 500 trauma patients [29]. As opposed to mechanical transduction to sense the viscoelastic parameters of clot formation, the TEG 6s measures changes in resonant frequency of a 1 mL blood samples using an automated cartridge system (thus avoiding errors associated with pipetting by hand as in the earlier TEG 5000 system). As the blood clots, the change in resonant frequency can be used to measure changes in viscoelasticity, generating a tracing and parameters that are conceptually similar, although not directly comparable, to the TEG 5000 legacy cup-and-pin system [30,31].

### 1.2. Rotational Thromboelastometry (ROTEM)

TEG and ROTEM have similar fundamentals, but in ROTEM, the pin oscillates while the cup remains stationary. The ROTEM tracing is like that of the TEG, but the parameters’ nomenclature and timing of measurement are different (see Figure 1). R coincides with the clotting time (CT), K represents the clot formation time (CFT), and MA represents the maximum clot firmness (MCF). Both analyzers use the alpha angle. ROTEM has multiple parameters to describe fibrinolysis: maximum lysis (ML) refers to the percentage decrease in amplitude relative to the MCF at the end of the run, while the clot lysis index (CLI/CLY/LI) is the ratio of the amplitude of the curve at 30/60 min after reaching MCF [32]. Multiple studies have described the applicability of TEG and ROTEM in cardiac surgery, liver transplantation, trauma resuscitation, and postpartum hemorrhage. There is little disparity between these two tests in the guidance of treatment [20,33,34,35,36,37,38,39].

The ROTEM Sigma is a fully automated test that is more convenient and easier to use than the previous Delta iteration; it provides a cartridge-based system and automated pipetting, but relies on the same cup-and-pin technology used in the ROTEM Delta. Studies comparing the ROTEM Sigma to its Delta predecessor have raised some questions regarding the validity of the reference ranges of the ROTEM Sigma [40,41,42,43].

### 1.3. Comparative Interpretation of Principle Viscoelastic Hemostatic Assays

In this section, our goal is to compare TEG and ROTEM in a way that will help clinicians better understand both analyzers. TEG and ROTEM have different terminologies, use different activators, and rely on both specific parameter values and the overall pattern recognition for interpretation. This reliance on assay familiarity and pattern recognition for bedside interpretation may be a barrier to a more common utilization of VHAs in orthopedics [30,39,44,45].

Rapid thromboelastography (rTEG) and the EXTEM assay of ROTEM both use an extrinsic cascade-targeted activation reagent, such as tissue factor, to initiate clot formation; thus, rTEG MA corresponds to EXTEM MCF [46]. Similarly, the kaolin TEG and the INTEM assay of ROTEM both use an intrinsic cascade-targeted activation reagent such as kaolin and ellagic acid. Both assays generate parameters describing the clot amplitude at 5 min intervals after R/CT until the graph reaches MA/MCF. These parameters are A5, A10, etc., in rTEG and CA5, CA10, etc., in EXTEM [44]. These values, in addition to the five key parameters described above, allow the real-time assessment of clotting abnormalities, allowing for an early analysis of hemostatic competence and anticipation of necessary blood product therapy (Table 1) [40,45,47,48]. Similarly, the functional fibrinogen MA of TEG and the FIBTEM assay of ROTEM both activate clot formation in the presence of platelet inhibitors, producing a tracing reflective of fibrin-specific contributions to clot formation and allowing the prompt recognition of a deficiency in fibrinogen [49].

A useful analogy for the interpretation of TEG/ROTEM tracings is to compare them to the shape of three shovels (Figure 2). For hemostatic competence or physiologic hemostasis, shovel-shaped tracing has an ideal handle length, blade slope, blade width, and blade tip, shown as the top shovel in Figure 2. The interpretation of the shovel shape and correspondence to ROTEM/TEG parameters is as follows: the CT/R is the length of the handle, the alpha angle is the slope of the blade, MCF/MA is the width of the blade, and CLI60/LY30 is the taper or point of the blade. The extremes of the coagulation spectrum are represented by different shovels, where, for the sake of analogy, the ease of tilling and moving soil corresponds to the ease of moving blood. The hypercoagulable shovel tracing has a short handle and a wide blade with an absent tip (middle shovel in Figure 2), which makes it very difficult for the earth to be broken up and transported. The hypocoagulable shovel tracing has a long handle with a narrow and pointed blade (third shovel in Figure 2), which makes tilling and soil transport less cumbersome but markedly inefficient. In the above examples, by analogy, tilling and soil transport are either difficult (hypercoagulable) or easy but inefficient (hypocoagulable). In summary, in Figure 2, the first shovel represents hemostatic equilibrium, the second shovel represents hypercoagulable disequilibrium, and the third shovel represents hypocoagulable disequilibrium. Goal-directed therapy can be given based on either the corresponding shovel-analogous pattern or the values of the various TEG and ROTEM parameters (Table 1).

### 1.4. Newer Viscoelastic Hemostatic Assays in Orthopedic Patients

The Quantra QPlus System (manufacturer: HemoSonics, LLC; Charlottesville, VA, USA) is a recently developed device approved for use in cardiac and orthopedic surgery which uses ultrasound-induced resonance to assess the dynamic properties of clot formation. The use of Quantra QPlus has been studied in a comparative analysis between Quantra QPlus, ROTEM parameters, and CCTs in a prospective observational study of 277 patients undergoing orthopedic surgery (79 orthopedic patients) or cardiac surgery. Good accuracy and strong inter-parameter correlations were seen between the ROTEM delta and Quantra. The results from receiver-operator curve analysis uncovered specificities and sensitivities in the 80–90% range when Quantra Qplus parameters were used to discriminate ROTEM threshold values commonly used in algorithms for goal-directed BCT for orthopedic patients [50]. With specific regard to coagulation factors such as cryoprecipitate and soluble fibrinogen concentrate, recent studies using Quantra QPlus have shown that fibrin and platelet contributions to clot stiffness measured by the device demonstrated thresholds that allowed for a laboratory-based fibrinogen and platelet threshold assay to guide transfusion decisions in orthopedic patients. Cutoff values were demonstrated for the lowest fibrinogen and platelet thresholds that would trigger the transfusion of cryoprecipitate, soluble fibrinogen, or platelet concentrates [51].

In addition, a new handheld VHA called the viscoelastic coagulation monitoring system (VCM: Entegrion; Durham, NC, USA) operates based on standard thromboelastography principles, but is significantly more portable than either TEG instrument. It has been approved for clinical use in Europe. In one study of patients undergoing orthopedic surgery, VCM parameters were compared to ROTEM delta and NATEM parameters. Specifically, the CT, CFT, alpha angle, A10, A20, MCF and lysis results showed a good correlation between the VCT system and ROTEM and NATEM [52].

As clinical experience with newer, optimized, and more portable VHA devices continues to grow, their clinical integration and optimal usage compared to TEG and ROTEM will be better elucidated.

## 2. Viscoelastic Hemostatic Assays in Orthopedic Surgery

### 2.1. Trauma

#### 2.1.1. Pelvic Fractures

There is a sparsity of literature specifically addressing the utilization of VHAs for the guidance of resuscitation in patients with pelvic fractures, which is surprising as this is a quintessential orthopedic injury pattern known to be associated with significant hemorrhage-related morbidity and mortality (Table 2). The high incidence of massive transfusion associated with pelvic fracture has been considered a target area for increased VHA use to guide BCT/HAT and has yielded promising results for improving mortality in polytrauma patients with severe hemorrhage [6,8,13]. Despite literature highlighting the benefit of VHAs to treat massive hemorrhage, there are only four retrospective studies analyzing a total of 522 patients with pelvic fractures and high injury severity scores (Table 2) [22,23,53].

A retrospective study of VHA parameters measured at presentation in a group of 141 patients with traumatic pelvic and or acetabular fracture demonstrated a significant relationship between LY30 on admission, increased packed red blood cells transfused, and increased mortality, suggesting that aggressive BCT/HAT is indicated in pelvic fracture patients with significant fibrinolysis [53]. An additional retrospective review of 131 patients with pelvic trauma revealed that abnormal activated clotting times on admission rTEG was an independent risk factor of death. Specifically, the death rate was 52% in patients with rTEG ACT values >6 min. There was no significant association between CCTs and mortality [54]. These studies highlight the role of VHA in managing traumatic hypocoagulopathy.

However, trauma patients with pelvic fractures often present with hypercoagulability and impaired fibrinolysis (termed “fibrinolytic shutdown”), and less commonly with profound, life-threatening hypocoagulopathy and/or hyperfibrinolysis (HF) [55,56]. In a retrospective study of 210 patients by Nelson et al., 59% of pelvic trauma patients presented fibrinolytic shutdown based upon admission rTEG analysis. The overall incidence of VTE in this group was 11%. It was noted that patients undergoing pelvic external fixation had higher rates of VTE, but details of conversion surgical procedures and pelvic fracture patterns were not included [23].

VHA is an appealing tool for balancing the resuscitation of both hypo- and hypercoagulopathy in this patient population. A pilot study analyzing the use of platelet mapping (TEG/PM, a modification of the standard TEG assay which evaluates platelet responsiveness) suggested that including TEG/PM may facilitate a reduction in platelet usage and a reduced cost of resuscitation. In this study, the average ratio of packed red blood cells (pRBC): fresh frozen plasma (FFP): platelets was 2.5:1:2.8. This work supports the validity of the so-called “platelet first” resuscitation for those orthopedic trauma patients who require massive transfusion, and further suggests that this may be optimally guided by TEG/PM. This concept is based on military as well as civilian data, which have shown improved mortality for those massive transfusion patients who were given a higher ratio of platelets to packed cells [22,57,58,59].

#### 2.1.2. Long Bone Fractures

Isolated long bone fractures, similar to isolated pelvic fractures, may be associated with abnormalities across the hemostatic spectrum such that these patients can present with both a hypercoagulable state as well as hemorrhage, placing them at risk of both bleeding and early pulmonary emboli after resuscitation [60] (Table 3). One prospective observational study of 250 patients with femoral neck fractures who underwent repair with dynamic hip screw (DHS) or hemiarthroplasty demonstrated no statistical difference between the coagulation and fibrinolysis parameters between patients undergoing hemiarthroplasty and DHS insertion. In addition, no coagulation indices differed between spinal and general anesthesia. However, individual analysis of TEG parameters showed significant and persistent hypercoagulable changes in the postoperative mean K and MA values, but no significant alteration in mean LY30 or LY60. In a subgroup of 70 patients, venography was performed either based on clinical suspicion for deep vein thrombosis (DVT; 20% confirmed) or empirically on postoperative days 7–10 (28% positive for DVT); hypercoagulability on postoperative days 1–7 was significantly associated with DVT [61].

TEG and CCTs have also been shown in pooled femur and humerus fractures. In these patients, hypercoagulopathic findings have been described with decreased R and K parameters as well as an increased MA and alpha angle within four hours after fracture and immediately prior to surgery [62]. Although limited, these studies demonstrate the ability of VHA to assess for hypercoagulability as a marker of early risk for VTE during fracture management. Further research is warranted to guide the timing, dosage, and duration of VTE prophylaxis beyond empiric strategies.

#### 2.1.3. VHA-Directed Management of Orthopedic Trauma Patients

The use of VHA during the resuscitation and operative management of orthopedic trauma patients has been shown to help identify and manage hypocoagulability in multiple studies (Table 4). The analysis of TEG MA and ROTEM MCF in 182 trauma patients evaluated in a prospective observational study revealed that early MAs were lower in patients with TIC and that these values for TEG MA and ROTEM MCF allowed differentiation between nontransfused patients, patients who received <10 units of packed RBCs, and patient who required more than 10 RBCs within the first 6 h of admission. In addition, platelet count and fibrinogen concentration revealed significant and superior correlations with the A10s of both TEG MA and the ROTEM MCF [47]. One prospective observational study compared rTEG and functional fibrinogen (FF-TEG) to kaolin TEG among 404 patients with a median ISS of 13. All TEG parameters, except the rTEG MA and the kaolin TEG MA, correlated significantly with mortality in orthopedic patients who had been transfused. The time from the initiation of the VHA assay to A5 and A10 was the lowest for rTEG and the TEG functional fibrinogen compared with the kaolin TEG. It was noted that rTEG A5 reduced the time to result by greater than 50% compared with the rTEG MA [63]. Taken together, these two studies suggest that the A5 and A10 for both the TEG MA and the ROTEM MCF may allow a faster assessment of the orthopedic patient’s hemostatic integrity than waiting for a completed TEG MA and ROTEM MCF. In a large study of 402 patients with trauma injury severity score (ISS) >15, including those with orthopedic injury, 132 patients at admission demonstrated hyperfibrinolysis (HF), as measured by FIBTEM. Using a multivariable Cox regression analysis, FIBTEM-HF at admission was independently associated with inpatient mortality (29/132 patients expired, 22%), highlighting the important role of VHA in the early detection and treatment of fibrinolysis—a state which is otherwise invisible to CCTs alone [64].

Importantly, although VHA detected hypocoagulability and can guide BCT and/or HAT, it also identifies hypercoagulability, which may influence thromboprophylaxis management (Table 4). Tsantes et al. studied preoperative and postoperative ROTEM values for 198 patients with fractured hips compared to age-matched controls. There was no detectable difference between the fracture and control patients based on CCTs. On the other hand, the ROTEM EXTEM parameters CT, CFT, MCF, and alpha angle, as well as the INTEM parameters CT, CFT, A10, MCF, and alpha angle, were significantly hypercoagulable in orthopedic fracture patients compared to age-matched controls. In addition, INTEM CT and CFT significantly decreased, while INTEM MCF, A10, and alpha angle significantly increased postoperatively, reflecting the exacerbation of existing hypercoagulability by fracture fixation. Thus, a hypercoagulable state following hip fractures and their operative repair can be detected by ROTEM soon after the initial trauma, which is not uncovered by conventional coagulation assays [65].

Among patient identifiable risk factors associated with hypercoagulability as determined by VHAs, age and gender have been specifically analyzed within the orthopedic population. In a study of patients undergoing orthopedic surgery for fracture repair, age was weakly correlated with increased hypercoagulability with all TEG parameters studied. This correlation was stronger for men than for women, and only R was significantly correlated with age when women were separately analyzed. In optimized regression models, age remained an independent predictor of hypercoagulable R, K, and alpha parameters. Therefore, age may need to be taken into account when interpreting TEG parameters and managing hypercoagulability in the elderly orthopedic trauma population [66].

### 2.2. Viscoelastic Hemostatic Assays in the Orthopedic Subspecialties

#### 2.2.1. Arthroplasty

The diagnosis of coagulation abnormalities and their perioperative management during elective arthroplasty is also facilitated by the use of VHAs (Table 5). With respect to the use of VHAs to prospectively identify hypocoagulability as a risk factor for bleeding complications, one 75-patient observational study evaluating ecchymosis after TKA identified that a hypercoagulable alpha angle and coagulation index (CI: a calculated summary parameter taking into account TEG R, K, angle, and MA) were protective against the incidence of postoperative ecchymosis, which occurred in 25/75 (33%) patients [62]. In this group, TEG CI was an important prognostic tool for predicting operative blood loss, and the change in the TEG CI allowed the prediction of significant blood loss and the development of ecchymosis after TKA [67].

However, similar to orthopedic trauma scenarios, hypercoagulability was markedly common after arthroplasty. In one study, 20 patients underwent general anesthesia versus 32 receiving spinal anesthesia for THA. TEG demonstrated a progressively increased hypercoagulable MA postoperatively. The TEG also demonstrated intraoperative hypercoagulability that was temporary in patients receiving spinal anesthesia. As expected and reported by others, CCTs were normal in all patients in the pre- and postoperative periods except for an increase in fibrinogen concentration at day five postoperation. Despite VTE prophylaxis, patients following THA are in a hypercoagulable state, as measured by both TEG and by CCT fibrinogen measurement. It is proposed that these patients may benefit from more optimal anticoagulation and or additional perioperative hemostatic monitoring using VHAs [68].

The effect of thromboprophylaxis to mitigate this hypercoagulability has also been investigated using VHAs (Table 5). One study identified 90 patients undergoing elective unilateral arthroplasty surgery of the hip or knee followed by 10-day enoxaparin thromboprophylaxis. At day 9 postoperation, 34/90 (37.8%) of patients were found to be hypercoagulable. Most patients (26/34 patients, 76.5%) demonstrated a mixed hypercoagulable state with features of platelet hyperactivity. It was postulated that serial TEG analysis allowed the identification of hypercoagulopathic patients at risk for VTE following total hip and knee replacement, and that platelet hyperactivity may represent a specifically important factor in orthopedic patients’ hypercoagulable state [69].

The use of antifibrinolytic tranexamic acid (TXA) has been suggested to contribute to the hypercoagulable perioperative state in elective orthopedic surgery. One study evaluated the effect of TXA in primary hip arthroplasty (THA) and TKA, finding that multiple doses of TXA contributed to a hypercoagulable state, as demonstrated by shortened R times on postoperative days one and three for both THA and TKA. CCTs, as expected in cases of hypercoagulability, demonstrated no change. However, this prothrombotic state did not contribute to an increased incidence of VTE when patients were provided appropriate thromboprophylaxis [70]. On the other hand, a similar study of the effect of TXA demonstrated significant reductions in total blood loss, units of packed red blood cell transfusion, and overall transfusion rates without an increase in VTE complications. It was found that TEG analysis did not change with the administration of TXA in this group of patients. Specifically, 86 patients received a single dose of TXA in the standard 15 milligrams per kilogram IV dose compared to 88 controls without TXA, and TEG parameters and CCTs were collected on the day prior to surgery and on the first and seventh postoperative days. There was no difference in any TEG parameters and CCTs between the two groups. However, total blood loss and drain blood loss in the TXA group were significantly lower than in the control arthroplasty group [71]. A small study of 23 patients demonstrated that TXA was associated with a selective increase in inflammatory markers following TKA. Specifically, in the small group of 23 patients of whom 12 received TXA before and after surgery, the levels of the inflammatory mediators monocyte chemotactic protein (MCP)-1, tumor necrosis factor (TNF)-α, interleukin (IL)-1β and IL-6 significantly increased compared to non-TXA patients, which was further amplified postoperatively. The authors also describe little change in the viscoelastic clot strength or fibrinolysis caused by TXA in this small group of patients [72]. Overall, additional work remains to be conducted to fully characterize the role and mechanisms of TXA with respect to coagulation and fibrinolysis in orthopedic patients, but VHA remains a useful tool to identify and manage any potential TXA-associated hypercoagulability.

VHA allows for sensitive study of other nonhemostatic aspects of intraoperative conduct on coagulation as well (Table 5). In one study of 22 patients, VHAs were utilized to investigate the effect of surgical pain on coagulation and fibrinolysis related to tourniquet use during TKA. TEG MA levels were noted to rise after tourniquet inflation only in patients undergoing general anesthesia; however, after deflation, MA values in both general and epidural anesthesia increased significantly. Fibrinolysis parameters did not change in either group during tourniquet inflation, but increased by 160% in both groups after touniquet deflation. This study suggests that surgical or tourniquet pain or both may impact coagulation, and epidural anesthesia was a useful technique to prevent hypercoagulopathy [73]. Topcu et al. studied the TEG effects of different resuscitative solutions comprising either Ringer’s lactate, 6% hydroxyethyl starch, or 4% succinylated gelatin solutions on the perioperative coagulability of orthopedic patients. This study demonstrated that all three solutions changed the coagulability in all patients, but these coagulation changes remained within normal limits [74]. However, it is concerning that overzealous intraoperative nonblood fluid management can contribute to potential perioperative coagulopathies with the most concerning being a dilutional hypocoagulopathy in the bleeding orthopedic surgical patient.

Importantly for implementation, variability between venous and arterial samples is known for several frequently monitored resuscitation markers, such as lactate. Venous and arterial variability in VHA parameters was studied using the INTEM, EXTEM, and FIBTEM assays of ROTEM in 52 patients at the time of arterial line insertion, intraoperatively, and postoperatively. There was no change in any parameters between venous and arterial samples, demonstrating that VHA assessment was consistent across both venous and arterial blood sampling [75].

#### 2.2.2. Spine Surgery

The diagnosis of coagulation abnormalities and their perioperative management during elective spine surgery is also facilitated by the use of VHAs (Table 6). Specifically, a group of 80 patients undergoing lumbar spine fusion surgery were divided into aspirin-treated and aspirin-naïve groups, and analyzed via TEG. Correlation analysis showed no significant differences between aspirin-treated and aspirin-naïve groups in any assayed TEG parameter. This absence of dysfunction in aspirin-treated patients undergoing fusion suggests that the relaxation of the restriction of aspirin therapy to approximately 2–3 days prior to surgery rather than the standard seven days may be safe from a bleeding perspective [76].

In a prospective cohort study evaluating the effect of different resuscitative fluids, the hemostatic competence of 66 spine fusion patients randomly assigned to receive gelatin solution, hydroxyethyl starch, or Ringer’s lactate solution was assessed. Plasma obtained prior to the induction of anesthesia and at the end of surgery was assayed for markers of fibrinolysis, and exposed to recombinant tissue plasminogen activator (tPA) in vitro followed by modified ROTEM analysis to sensitively assay for fibrin clot strength. While markers of fibrinolytic activity did not differ between groups, the tPA-modified ROTEM assay revealed more tPA-resistant clot formation in patients managed with crystalloid as opposed to colloid fluids, suggesting a subtle coagulopathy related to clot instability in patients receiving colloids [77]. In a similar study by the same researchers using the same patient population, modified ROTEM was used to correlate the changes in alpha angle, clot firmness, and fibrinogen polymerization, as manifested by the FIBTEM assay. The FIBTEM alpha angle and MCF were decreased in patients receiving hydroxyethyl starch, followed by lower magnitude decreases with gelatin, and the least reduction seen with Ringer’s lactate. In addition, 13 patients in the colloid groups but none in the crystalloid group required fibrinogen concentrate to maintain borderline FIBTEM clot firmness. Further, the activities of FVII, FVIII, FIX, and von Willebrand factor activity were significantly decreased in the colloid groups. The authors concluded that intraoperative dilutional coagulopathy is largely a function of decreased fibrin polymerization, which may be exacerbated by colloid administration compared to crystalloids [78].

Because high perioperative blood loss has been noted in spine and spinal deformity surgery along with other areas of orthopedic surgery, methods to reduce intraoperative bleeding include the use of cell savage, new surgical techniques, the substitution of coagulation factors, antifibrinolytics, desmopressin, induced hypertension, and the avoidance of hypothermia. The use of VHAs to guide anticoagulation in the perioperative period has been suggested and thoroughly reviewed elsewhere by Oliver Theusinger and Donat Spahn [79].

#### 2.2.3. VHA-Guided Operative Conduct in Elective Orthopedic Surgery

The use of VHA to guide BCT and other elements of operative conduct has been shown to help identify and manage hypocoagulability during elective orthopedic surgery, similar to its utility in orthopedic trauma (Table 7). One large retrospective study evaluated the utility of TEG-guided BCT compared to CCT-guided BCT among 480 patients undergoing orthopedic surgery, finding that TEG-guided BCT was associated with lower overall transfusion volumes of RBC, FFP, platelets, and cryoprecipitate. In addition, the TEG-guided BCT patients were found to have shorter hospital stays compared to the CCT-guided group. This suggests that the TEG guidance of intraoperative BCT during elective orthopedic surgery is associated with optimized blood product usage and potentially improved outcomes overall [80]. The ability of CCTs versus ROTEM to identify perioperative coagulopathy in patients who underwent major elective orthopedic surgery was evaluated in 40 patients undergoing elective THA, femur fracture fixation, and spinal surgery. ROTEM analysis was compared to routine CCTs and thrombin generation profiles. Their findings demonstrated a correlation between a lower platelet count and ROTEM MCF in bleeding patients. At baseline, a significantly lower platelet count and FIBTEM MCF were observed in patents with excessive bleeding. Thrombocytopenia and a low fibrinogen activity were also associated with intraoperative bleeding in these patients. It was therefore proposed that ROTEM identified hypocoagulopathic phenotypes in patients receiving major elective orthopedic surgery [81]. In a prospective observational trial, perioperative coagulation monitoring and transfusion data were analyzed for 70 patients undergoing general orthopedic surgery. Of the 23 patients who were transfused, there was a reduction in FIBTEM below the 7 mm threshold. In addition, preoperative CT was also prolonged in the transfusion group when compared to the nontransfused group, identifying ROTEM parameters predictive of and associated with bleeding that may serve as useful targets to optimize BCT [82].

One study specifically evaluated the effect of intraoperative autotransfusion via cell salvage in 25 hip arthroplasty patients. ROTEM analysis as well as CCTs and cytokine levels were evaluated prior to surgery and within one hour of reinfusion of 300 cc or more of salvaged whole blood (average of 460 cc). ROTEM analysis demonstrated normal clot formation at baseline and no parameter significantly changed following reinfusion. In addition, CCTs were all normal in these patients. However, MCP-1 but no other cytokine levels were elevated after reinfusion. Therefore, the reinfusion of salvaged whole blood did not appear to alter the hemostatic integrity of these patients when assayed by either ROTEM or CCTs [83].

Similarly to the findings on the management of orthopedic trauma identified above (Table 4), the use of VHA during elective orthopedic surgery may allow for the optimization of nonhemostatic aspects of intraoperative conduct as well (Table 7). For example, one study evaluated the effect of various anesthetic and paralyzing agents on hemostatic integrity, identifying that supratherapeutic doses of sugammadex were associated with reduced clot strength as determined by TEG [84].

### 2.3. Prediction and Prevention of Venous Thromboembolism

The area where VHAs have found the greatest utility in the field of orthopedics is in the prediction and prevention of VTE. In a major study of 349 patients at eight level 1 European trauma centers, VTE incidence was found in 58% of patients sustaining major trauma who were screened by venography and who were not administered chemical thromboprophylaxis. Of 182 patients with pelvic and/or long bone lower extremity fractures, 69% developed VTE, with multivariate analysis demonstrating femur or tibia fracture as an independent risk factor for VTE [85]. Increased ISS scores have also demonstrated an increased odds ratio of developing VTE [86,87]. In addition, a delay in fracture fixation of four days or more post-trauma results in a three-fold increased risk of DVT in the trauma population [88]. Through their ability to identify and monitor hypercoagulability, VHAs offer a unique and singular opportunity to reduce the incidence of VTE postoperatively for orthopedic trauma patients (Table 8).

In a systematic review of 31 studies, 17 studies cited an elevated MA as a significant predictor of VTE among 6348 total patients. A subsequent subcohort meta-analysis comprising 3180 patients from five studies demonstrated the ability of an MA >65 mm to predict VTE. However, there were inconsistencies in the cut-off MA value to define hypercoagulability in those patients who developed VTE. Other studies, however, have demonstrated that an MA >65 mm is a useful threshold for VHA prediction in orthopedic trauma patients [89]. Specifically, a group of 1818 patients stratified by extremity abbreviated injury severity scores (AIS) of 2+ or <2 were shown to have an odds ratio of developing a VTE of 3.66 for an MA >65 mm and a ratio of 6.7 for an MA >72 mm [90]. The use of surveillance ultrasound in orthopedic trauma patients to determine the incidence of postoperative VTE has been studied in prospective observational studies. Specifically, 983 patients, of whom 684 received both an admission TEG and surveillance ultrasound during admission, revealed that a VTE was diagnosed in 14.5% (99/684) of these patients. Of these 684 patients who received both a surveillance ultrasound and a TEG at index admission, 582 or 85.1% demonstrated hypercoagulable findings. Those patients with hypercoagulable findings had a much higher rate of lower extremity VTE than those who did not have a hypercoagulopathic TEG (15.6% versus 8%). After adjustment for relevant covariates, multivariate analysis demonstrated that a hypercoagulable admission TEG was associated with VTE with an odds ratio of 2.41 [24]. An additional retrospective study over 13 months evaluated 354 patients with peritrochanteric and femoral neck fractures who underwent hemiarthroplasty or cephalomedullary nailing. ROTEM parameters were evaluated for correlation with the hypercoagulable state and symptomatic VTE. ROTEM analysis was performed within hours of injury and also on the second postoperative day. The diagnosis of VTE was made by CT pulmonary angiography or vascular ultrasound, which was performed only in symptomatic patients. No routine screening exams were performed. Specific ROTEM parameters were compared between symptomatic and asymptomatic VTE. Preoperative MCF was found to be higher and postoperative CFT lower in patients with subsequent clinically proven VTE than in those patients without VTE. Based on this study, ROTEM provided a high level of accuracy in the detection of postoperative hypercoagulability after orthopedic surgery that was predictive of later symptomatic VTE [91].

However, not all studies have identified as clear a causal link between hypercoagulability and subsequent VTE. A differential population-based study used TEGs to test the hypothesis and confirm the clinical observation that the incidence of VTE was lower in Asian patients than in the Western population. The incidence of VTE was studied in 101 patients who had THA or TKA or surgeries for hip fractures without chemoprophylaxis for DVT. TEG was adopted to predict the presence of a hypercoagulable state and predict the occurrence of VTE. The incidence of VTE in this population was 7%. Of the patients who developed VTE, 6/7 had hip fractures while one had a TKA. Preoperative TEG was hypercoagulable in only 1/7 patients with VTEs but was also hypercoagulable in 37 of the remaining 94 patients without VTE. The incidence of DVT in the study population was sufficiently high to recommend some form of DVT prophylaxis in this Asian population following hip and knee surgery. However, TEG assessment preoperatively did not predict the risk of VTE [92]. These results are supported by a recent meta-analysis including more than 8000 postoperative patients across several surgical disciplines, which identified common hypercoagulability in postoperative patients but did not find that TEG-based MA was a significant predictor of VTE [89]. While work remains to be conducted to identify specific VHA parameters and patient factors predictive of VTE in orthopedic patients, overall, VHA-based hypercoagulability appears to be a significant risk factor for VTE.

Many initial studies of TEG in orthopedic surgery involved the use of TEG to monitor anticoagulant effects. In a group of 52 patients undergoing TKA who received prophylactic warfarin and epidural anesthesia, daily INR values and TEG parameters were compared with preoperative values. The threshold for the removal of the epidural catheter was on POD2 if the INR was ≤2. If the INR was persistently elevated >2 on POD2, warfarin was held until the INR dropped ≤2 prior to catheter removal. On the day of epidural catheter removal, the INR was elevated (mean INR, 1.5), while the TEG parameters K, MA, alpha angle, and CI remained normal in those patients who received low-dose warfarin DVT prophylaxis [93]. With respect to LMWH chemoprophylaxis, anti-Xa levels as a marker for enoxaparin efficacy were compared to TEG parameters for the ability to predict VTE prophylaxis effectiveness. In a prospective study of 24 patients who had unilateral TKA or THA, the mean R time and K time in TEG correlated with the anticipated peak and trough levels of enoxaparin and serum anti-Xa levels. It was shown that the R time and K time on postoperative day 3 indicated an exaggerated response to enoxaparin, and TEG was suggested as a test that could potentially monitor the degree of anticoagulation required to provide effective VTE prophylaxis for patients administered low-molecular-weight heparins [94].

The use of VHAs to direct VTE prophylaxis has traditionally involved the use of LMWH for orthopedic patients. However, recent evidence suggests that novel oral anticoagulants (NOACs, including direct thrombin inhibitors and Factor Xa inhibitors) may be efficacious and more cost-effective than LMWH for VTE prophylaxis after orthopedic surgery [95]. Whether VHAs can be used to monitor NOAC-based prophylaxis is an active area of investigation. One proof-of-concept study investigated the in vitro effect of varying concentrations of dabigatran (a direct thrombin inhibitor) on whole blood, demonstrating that dabigatran prolonged the TEG R time, and correlated well with validated thrombin inhibitor tests such as the Hemoclot thrombin inhibtor assay (manufacturer: HYPHEN BioMed; Neuville-sur-Oise, France) and ecarin clotting time [96]. Another proof-of-concept study including the Xa inhibitors rivaroxaban and apixaban in addition to dabigatran showed that the in vitro addition of these agents to healthy volunteer blood samples led to the prolongation of both the R time and the maximum rate of thrombin generation, demonstrating the feasibility of monitoring NOAC therapy with TEG [97]. In a prospective observational study of 188 patients undergoing elective THA or TKA who were administered either a 40 mg daily subcutaneous dose of enoxaparin or 10 mg oral dose of rivaroxaban, pre- and postoperative ROTEM parameters were analyzed and compared to thrombin generation and coagulation activation markers. Rivaroxaban increased the extrinsic ROTEM CT more robustly than enoxaparin, and resulted in impaired clot initiation and propagation in thrombin generation studies as well as lower plasma markers of clot formation. Interestingly, the levels of the anticoagulant antithrombin III were reduced after enoxparin treatment but preserved after rivaroxaban treatment, suggesting the preservation of natural anticoagulants as a mechanism for more efficacious VTE prevention compared to enoxaparin [98].

### 2.4. Future Study

The current literature surrounding the use of VHAs in orthopedic surgery has provided a foundation, but there are gaps in the literature that need to be addressed. There remains a necessity for longitudinal studies following orthopedic patient outcomes with and without the use of VHAs to guide management. Additionally, studies to assess the prognostic value of VHAs in both elective and nonelective orthopedic surgery patients may allow for proactive and beneficial changes in patient management.

### 2.5. Summary of Principles for VHA Use in Orthopedic Patients

This review has summarized data supporting three key assertions:

VHAs optimize the resuscitation and intraoperative management of hypocoagulability and hyperfibrinolysis in both orthopedic trauma and elective surgical patients.

VHAs also detect clinically significant hypercoagulability and impaired fibrinolysis, which are otherwise undetectable by CCTs.

The VHA-based identification and management of hypercoagulability has significant implications for risk stratification and chemoprophylaxis management of VTE prophylaxis in orthopedic patients.

## 3. Conclusions

Orthopedic coagulopathy consists of a spectrum of phenotypes that are multifactorial in relation to patient factors, injury patterns, the conduct of resuscitation and surgical procedures, and perioperative VTE prophylaxis strategies. The last 20 years have seen a dramatic growth of bedside VHA use for the management of both hypo- and hypercoagulopathies, with orthopedic surgery being relatively late to the game. There has been an expansion of the routine use of these tests in the management of not just trauma, but also in all postsurgical, obstetrical, and medical intensive care units where severe bleeding is often encountered. Since major hemorrhage is not an uncommon finding in orthopedic trauma, the same rigor of monitoring these patients at the bedside during the initial stages of resuscitation now requires that orthopedic traumatologists adapt and familiarize themselves with the use of VHAs. Beyond orthopedic trauma alone, there is also a profound proof of concept that VTE prophylaxis should entail a VHA-driven individualized approach for dosing and duration as compared to the antiquated empiric strategy. The future use of VHAs in orthopedic trauma and elective surgery must become more commonplace. It is imperative that current and future orthopedic surgeons are educated on the applied science, rationale, and methodical use of VHAs as they relate to BCT transfusions, the administration of HAT products, the pre-operative assessment and management of patients taking anticoagulants, and the management of VTE prophylaxis. Future studies are necessary to further expand the routine acceptance and use of VHAs over CCTs, which will only begin with a widespread acceptance of VHAs by the orthopedic surgical community.

## Figures and Tables

**Figure 1 jcm-11-04029-f001:**
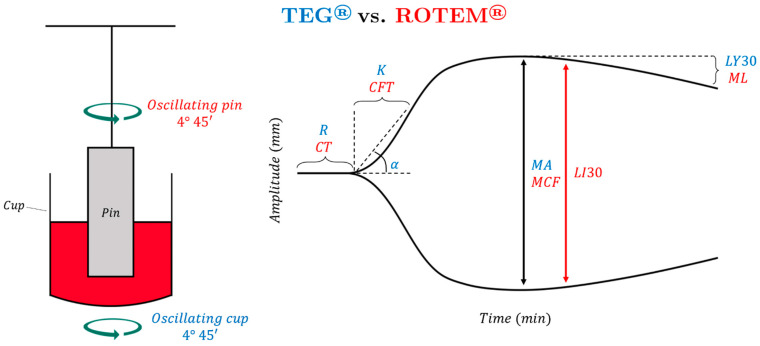
A 0.36 mL citrated sample of whole blood is placed into a heated cup maintained at 37 °C in which a pin is suspended. In TEG, the cup oscillates while the pin is stationary; in ROTEM, the pin oscillates while the cup remains stationary. In both assays, citrate anticoagulation is reversed with calcium, and a coagulation activation reagent is added; as the blood coagulates, force is detected by the pin and transmitted to an electrical transducer, producing a graphical output that represents clot dynamics over time. Reaction time (R) and clotting time (CT) represent the amount of time until the tracing reaches 2 mm in amplitude. Kinetics (K) and clot formation time (CFT) represent the amount of time between 2 mm of amplitude and when the tracing reaches 20 mm in amplitude. The α-angle is used in both TEG and ROTEM; it represents the angle between the horizontal line and the line between clot initiation and 20 mm amplitude. Maximum amplitude (MA) and maximum clot firmness (MCF) represent the maximum amplitude that the tracing reaches. Lysis at 30 min (LY30) represents the percentage decrease in amplitude 30 min after reaching MA. Clot lysis index (CLI30) represents the percentage of amplitude remaining 30 min after reaching MA. Maximum lysis (ML) refers to the percentage decrease in amplitude relative to the MCF at the end of the run.

**Figure 2 jcm-11-04029-f002:**
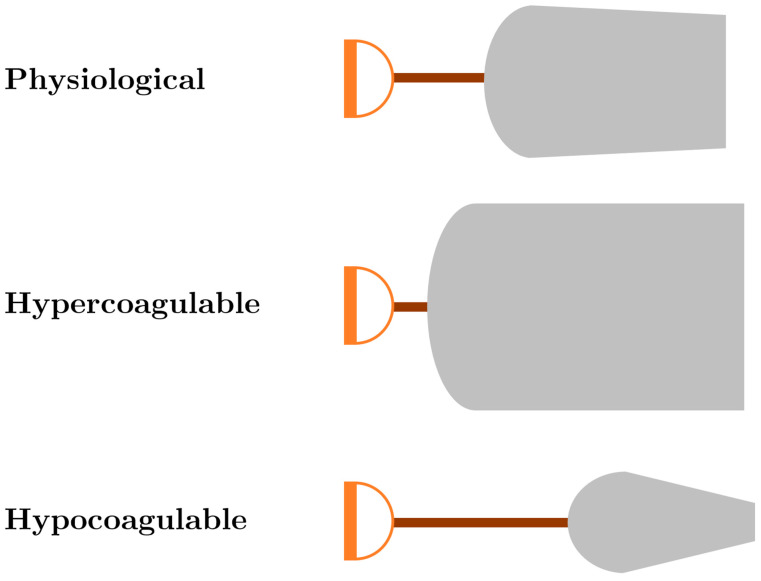
Shovel analogy of ROTEM/TEG tracings. The first shovel, shown with a normal handle length and moderately wide shovel blade with a slight narrowing of the tip of the blade, represents physiological hemostasis with normal CT/R, alpha angle, MCF/MA, and CLI60/LY30. The second ROTEM/TEG shovel tracing with a very short handle and wide blade with very little tapering of the tip of the blade depicts a tracing with a decreased CT/R and a wide alpha angle, CT/MA, and increased width of the CLI60 with very low LY30. This shovel tracing is indicative of a hypercoagulable/fibrinolytic shutdown phenotype. The third ROTEM/TEG shovel tracing depicts a tracing with a prolonged CT/R, narrow alpha angle, a narrow MCF/MA, and increased lysis represented by the decreased width of CLI60 and the pointed and increased LY30; these parameters are indicative of a hypocoagulopathic/hyperfibrinolytic phenotype. See Table 1 for a treatment algorithm based on this shovel analogy. Abbreviations: clot lysis index at 30/60 min (CLI30/CLI60); clotting time (CT); lysis at 30 min (LY30); maximum amplitude (MA); reaction time (R); rotational thromboelastometry (ROTEM); thromboelastography (TEG).

**Table 1 jcm-11-04029-t001:** Simplified algorithm of blood component therapy and hemostatic adjuncts based on the shovel analogy for viscoelastic hemostatic assays [8,14].

ROTEM/TEG Parameter Abnormality	Intervention
Prolonged CT/R, “Long handle”	Fresh frozen plasma and/or multifactor concentrates (e.g., prothrombin complex concentrate)
Decreased CT/R “Short handle”	Prophylactic anticoagulation
Decreased alpha angle, “Decreased slope of blade”	Cryoprecipitate/fibrinogen concentrate
Decreased MCF/MA, “Decreased width of blade”	Platelets/cryoprecipitate/fibrinogen concentrate
Increased MCF/MA “Increased width of blade”	Prophylactic anticoagulation and/or antiplatelet agents
Increased CLI60/LY30, “Sharp point of blade”	Antifibrinolytic (e.g., tranexamic acid)

Abbreviations: clot lysis index at 60 min (CLI60); clotting time (CT); lysis at 30 min (LY30); maximum amplitude (MA); maximum clot firmness (MCF); reaction time (R).

**Table 2 jcm-11-04029-t002:** Summary of the literature pertaining to viscoelastic hemostatic assays applied for the treatment of patients with pelvic fractures.

First Author	Study Design	No. of Patients	Conclusions
Bostian et al., 2020 [53]	Retrospective Cohort	141 pelvic and/or acetabular fractures	Increased LY30 on index TEG significantly correlated with mortality, blood loss, and pRBCs transfused.
Kane et al., 2015 [54]	Retrospective Cohort	131 pelvic and/or acetabular fractures	Index TEG R >6 was an independent predictor of mortality (13/25, 52% death rate). Forty-one patients had an abnormal R value at index presentation.
Mamczak et al., 2016 [22]	Retrospective Cohort	40 pelvic and/or acetabular fractures	TEG/PM-guided resuscitation yielded an average transfusion ratio of 2.5:1:2.8 pRBC:FFP:platelets. TEG may optimize resuscitation over standard 1:1:1 fixed ratio guidelines.
Nelson et al., 2020 [23]	Retrospective Cohort	210 severe pelvic fractures	At index presentation, 59% of patients demonstrated fibrinolytic shutdown on rTEG. VTE incidence was 11%, and fibrinolysis shutdown on rTEG at index did not predict VTE.

Abbreviations: fresh frozen plasma (FFP); lysis at 30 min (LY30); packed red blood cell (pRBC); rapid thromboelastography (rTEG); thromboelastography (TEG); thromboelastography with platelet mapping (TEG/PM); venous thromboembolism (VTE).

**Table 3 jcm-11-04029-t003:** Summary of the literature pertaining to viscoelastic hemostatic assays applied for the treatment of patients with long bone fractures.

First Author	Study Design	No. of Patients	Conclusions
Wilson et al., 2001 [61]	Prospective Cohort	250 femoral neck fractures	Patients who developed postoperative DVT demonstrated significantly greater hypercoagulability by TEG on PODs 1–7 compared to those who did not develop DVT.
Liu et al., 2016 [62]	Retrospective Cohort	40 fractures (13 humerus and 27 femur) in adults >60 years old, compared to 40 age-matched controls	Prior to surgery and 4 h post injury, aged fracture patients showed significant hypercoagulability on TEG by decreased K and R and increased alpha angle, MA, and CI compared to controls. TEG parameters correlated well with CCTs.

Abbreviations: common coagulation test (CCT); coagulation index (CI); deep vein thrombosis (DVT); clot kinetics (K); maximum amplitude (MA); postoperative day (POD); reaction time (R); thromboelastography (TEG).

**Table 4 jcm-11-04029-t004:** Summary of the literature pertaining to viscoelastic hemostatic assay-guided goal-directed therapy for orthopedic patients.

First Author	Study Design	No. of Patients	Conclusions
Meyer et al., 2014 [47]	Prospective Cohort	182 trauma patients	A5 and A10 may be better predictors of TIC and need for goal-directed transfusion compared to MA/MCF in severely injured trauma patients.
Laursen et al., 2018 [63]	Prospective Cohort	404 trauma patients	The following admission TEG parameters were found to predict mortality: kTEG A10 and MA; FF A5, A10, and MA. For transfused patients, all TEG parameters were predictive of mortality except for rTEG MA and kTEG MA.
Wang et al., 2020 [64]	Retrospective Cohort	402 trauma patients with ISS > 15	Hyperfibrinolysis on FIBTEM at index presentation was independently associated with a significantly higher mortality (22.3%) compared to those without hyperfibrinolysis (10.3%).
Tsantes et al., 2021 [65]	Case-Control Study	198 hip fractures undergoing arthroplasty or intramedullary nailing, age-matched to 52 controls	Post injury, hip fracture patients demonstrated hypercoagulability by abnormal EXTEM MCF and alpha angle, and INTEM MCF, A10, and alpha angle. Postoperative ROTEM analysis trended towards increased hypercoagulability
Ng 2004 [66]	Retrospective Cohort	132 fracture repair patients	Prior to anesthesia induction, age weakly correlated to increasing hypercoagulability on TEG.

Abbreviations: injury severity score (ISS); maximum rate of thrombin generation (MRTG); reaction time (R); rotational thromboelastometry (ROTEM); thromboelastography (TEG); time to maximum rate of thrombin generation (TMRTG).

**Table 5 jcm-11-04029-t005:** Summary of the literature pertaining to viscoelastic hemostatic assays applied to patients undergoing arthroplasty.

First Author	Study Design	No. of Patients	Conclusions
Wang et al., 2018 [67]	Prospective Cohort	75 unilateral TKA patients. All patients received 10 mg rivaroxaban postoperatively once daily.	Ecchymosis postoperatively was significantly associated with a postoperatively increased R and decreased alpha angle and CI compared to preoperative TEG analysis.
Lloyd-Donald et al., 2021 [68]	Retrospective Cohort	52 elective THA patients (20 general anesthesia, 32 spinal anesthesia). All patients received postoperative DVT chemoprophylaxis.	Regardless of anesthesia technique, postoperative TEG demonstrated hypercoagulability by gradually increased MA through POD5. CCTs only demonstrated significantly elevated fibrinogen levels postop for both anesthesia techniques. TEG/CCT correlation to VTE was not reported.
Yang et al., 2014 [69]	Prospective Cohort	90 TKA or THA patients. All received 10 days thromboprophylaxis with enoxaparin	On POD9, 37.8% of patients were hypercoagulable on TEG despite thromboprophylaxis. There was a significant trend towards hypercoagulability on PODs 1–4 for parameters K, MA, alpha angle, and CI. MA and alpha angle remained increased on PODs 4–9.
Wu et al., 2019 [70]	Retrospective Cohort	359 THA or TKA patients who received at least 1 dose of TXA immediately prior to surgery.	Patients who received multiple doses of TXA preoperatively demonstrated significantly greater hypercoagulability by TEG for 7 days postoperation compared to those who only received a single dose of TXA. However, using multiple doses of TXA was not correlated to VTE.
Grant et al., 2018 [72]	Prospective Cohort	23 TKA (12 received TXA preoperatively and immediately after surgery)	No significant difference in ROTEM parameters was detected for those who received TXA compared to those who did not receive TXA.
Zhang et al., 2020 [71]	Prospective Cohort	174 THA (86 received TXA, 88 no TXA)	There was no significant difference in coagulation status by TEG or CCT analysis on the day before operation, POD1, or POD7 for those patients who received TXA. There was no difference in VTE rate. The TXA group had significantly lower blood loss and transfusion requirement.
Kohro et al., 1998 [73]	Prospective Cohort	22 TKA patients (11 extradural anesthesia (EA) and 11 general anesthesia (GA))	Tourniquet inflation was associated with an increase in MA greater in the GA group. Fibrinolysis significantly increased in both anesthesia groups 5 min after tourniquet deflation.
Topcu et al., 2012 [74]	Prospective Cohort	75 orthopedic patients undergoing TKA or THA	Maintenance fluids of Ringer’s lactate, 6% hydroxyethyl starch, and 4% gelofusine solution each had mild changes on TEG parameters immediately and 24 h postoperation, but all changes were within normal limits.

Abbreviations: common coagulation test (CCT); clot formation time (CFT); coagulation index (CI); deep vein thrombosis (DVT); clot kinetics (K); maximum amplitude (MA); maximum clot formation (MCF); postoperative day (POD); reaction time (R); rotational thromboelastometry (ROTEM); thromboelastography (TEG); tranexamic acid (TXA); venous thromboembolism (VTE).

**Table 6 jcm-11-04029-t006:** Summary of the literature pertaining to viscoelastic hemostatic assays applied to patients undergoing spine surgery.

First Author	Study Design	No. of Patients	Conclusions
Li et al., 2020 [76]	Retrospective Cohort	80 lumbar fusion patients (39 aspirin-treated and 41 aspirin-naïve)	There was no significant difference in perioperative TEG values between the two treatment groups.
Mittermayr et al., 2007 [78]	Prospective Cohort	66 spinal fusion patients	Magnitude of MCF reduction is affected by the type of intraoperative maintenance fluid used. Colloids showed a greater change in MCF compared to crystalloid. FIBTEM MCF <7 mm predicted clinical bleeding and the need for fibrinogen replacement. Hydroxyethyl starch demonstrates the most significant decrease in fibrin polymerization.
Mittermayr et al., 2008 [77]	Prospective Cohort	66 spinal fusion patients	As determined by an in vitro tPA challenge test (invoked hyperfibrinolysis), patients administered intraoperative maintenance fluids with colloids demonstrated more rapid clot dissolution compared to those provided crystalloid.

Abbreviations: common coagulation test (CCT); fresh frozen plasma (FFP); maximum amplitude (MA); postoperative day (POD); packed red blood cells (pRBC); thromboelastography (TEG); venous thromboembolism (VTE).

**Table 7 jcm-11-04029-t007:** Summary of the literature pertaining to viscoelastic hemostatic assays applied to the intraoperative conduct of patients undergoing elective orthopedic surgery.

First Author	Study Design	No. of Patients	Conclusions
Zhang et al., 2021 [80]	Retrospective Cohort	480 orthopedic surgery patients (266 TEG-guided, 214 CCT-guided; 202 spinal surgeries, 180 fracture surgeries, and 98 TKA)	TEG guided intraoperative transfusion required lower volumes of pRBCs, FFP, cryoprecipitate, and platelets.
Spiezia et al., 2016 [81]	Retrospective Cohort	40 elective orthopedic patients (THA, femur fracture fixation, and spinal surgery) who had ≥250 mL/h blood loss intraoperatively	Intraoperative blood loss significantly correlated with preoperative low platelet count and low FIBTEM MCF, and postoperative low fibrinogen and platelet count, prolonged CFT, and decreased alpha angle and MCF
Hanke et al., 2020 [82]	Retrospective Cohort	70 THA, TKA, or spinal fusion patients (23 received transfusions)	CT was significantly prolonged and FIBTEM decreased preoperatively in the group that required postoperative transfusions.
Froessler et al., 2015 [83]	Prospective Cohort	25 orthopedic patients undergoing THA	ROTEM did not indicate coagulopathy after reinfusion of unwashed salvaged whole blood.
Lee et al., 2018 [84]	Prospective Cohort	14 healthy adults undergoing elective orthopedic surgery compared to controls	Increasing supratherapeutic concentrations of sugammadex were significantly associated with decreases in coagulation as manifested by prolongation in TEG R time, time to maximum rate of thrombus generation (TMRTG), and decreases in the alpha angle, MA, and maximum rate of thrombus generation (MRTG).

Abbreviations: reaction time (R); thromboelastography (TEG).

**Table 8 jcm-11-04029-t008:** Summary of the literature pertaining to viscoelastic hemostatic assays applied to the prediction and prevention of venous thromboembolism among orthopedic operative patients.

First Author	Study Design	No. of Patients	Conclusions
Cotton et al., 2012 [86]	Retrospective Cohort	2070 trauma activations (53 developed PE)	Elevated MA at admission was an independent predictor of PE. MA >65 had an odds ratio of 3.5, and MA >72 had an odds ratio of 5.8.
Gary et al., 2016 [90]	Retrospective Cohort	1818 (310 extremity AIS ≥ 2, 1508 extremity AIS < 2)	Admission rTEG MA predicted VTE in patients with severe extremity trauma with an OR = 3.66 for MA ≥65 and OR = 6.7 for MA ≥72.
Brill et al., 2017 [24]	Prospective Cohort	684 trauma patients who received surveillance u/s and admission TEG	Admission TEG demonstrated hypercoagulability in 582 (85.1%) patients. LE DVT was diagnosed in 99 (14.5%) patients. Despite prophylaxis, hypercoagulable TEG carried a two-fold risk of DVT (OR 2.41, 95% CI 1.11–5.24).
Tsantes et al., 2021 [91]	Retrospective Cohort	354 femoral neck and peritrochanteric fracture patients	Several abnormal ROTEM values were found predictive of VTE: Increased preop MCF (median 70 mm), decreased preop CFT (median 61 s), and decreased postop CFT (median 52 s). Preop CFT demonstrated the greatest prediction of VTE with sensitivity of 81% and specificity of 86%.
Parameswaran et al., 2016 [92]	Prospective Cohort	101 hip fracture or elective THA/TKA patients who did not receive postoperative chemoprophylaxis for DVT	DVT incidence was 7%. Only 1 patient with DVT demonstrated hypercoagulability on preoperative TEG. For those without DVT, 37/94 demonstrated hypercoagulability on preoperative TEG. No TEG parameter was predictive of DVT.
Brown et al., 2020 [89]	Systematic review (35 studies) & meta-analysis (5 studies)	8939 postoperative patients	MA >65 was not predictive of VTE (OR 1.31, 95% CI 0.74–2.34). There was wide variability across studies for the threshold MA value of hypercoagulability and the pooled mean threshold value was 66.7 mm. TEG consistently showed hypercoagulability on POD1.
Hepner et al., 2002 [93]	Prospective cohort	52 TKA patients receiving DVT prophylaxis with warfarin	On the day of epidural catheter removal, R was increased compared to preoperative values but still within normal range. CI was normal. Only INR was elevated to an average of 1.48 at the time of catheter removal.
Klein et al., 2000 [94]	Prospective Cohort	24 unilateral TKA/THA patients	TEG parameters R and K correlated well with anticipated peak and trough of postoperative LMWH and anti-Xa levels.
Oswald et al., 2015 [98]	Prospective Cohort	188 orthopaedic surgery patients (receiving 40 mg enoxaparin or 10 mg rivaroxaban)	Increase in EXTEM CT was greater with rivaroxaban than enoxaparin.

Abbreviations: amplitude at 5 min (A5); amplitude at 10 min (A10); clot formation time (CFT); coagulation index (CI); clotting time (CT); deep vein thrombosis (DVT); functional fibrinogin (FF); general anesthesia (GA); international normalized ratios (INR); clot kinetics (K); kaolin thromboelastography (k-TEG); low-molecular-weight heparin (LMWH); maximum amplitude (MA); maximum clot formation (MCF); pulmonary embolism (PE); postoperative day (POD); reaction time (R); rotational thromboelastometry (ROTEM) rapid thromboelastography (rTEG); trauma-induced coagulopathy (TIC); venous thromboembolism (VTE).

## Data Availability

Not applicable.

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
