# Peer review of "Viscoelastic Hemostatic Assays for Orthopedic Trauma and Elective Procedures"

_jcm, 2022, doi:10.3390/jcm11144029_

Round 1
Reviewer 1 Report
The review entitled "Viscoelastic Hemostatic Assays for Orthopaedic Trauma and Elective Procedures " is on the whole a well-written manuscript. The aim of the study is to assess the emerging use of VHAs to quickly and reliably analyze real-time coagulation and fibrinolytic status in both orthopaedic trauma and elective orthopaedic surgery, as there have been several recent studies which support it. The data however are more reliable in the case of bleeding. Since the authors also refer to the use of VHAs to identify hypercoagulability for VTE risk assessment and for the monitoring of VTE prophylaxis, I believe that they need to clarify that until now the data are insufficient to permit an individualized approach with regard to dosing and duration for VTE thromboprophylaxis. Another comment I would like to make is that the term “heparinoid” is not appropriate and it should be replaced by “LMWH”.
Reviewer 2 Report
This review describes the use of viscoelastic hemostatic assays in orthopedic patients, both for trauma and in elective surgery. The paper neatly summarizes the various technologies (TEG, ROTEM, newer techniques) and explains the differences among the two. The technology is well explained, and I like the analogy with the shovel. It subsequently describes the available literature in this specific medical field.
Viscoelastic hemostatic assays are known for being used in for example trauma, where the transfusion strategy (including tranexamic acid, clotting factors etc.) is guided by multiple measurements with the device. The current review focuses on orthopedic patients. It appears that the majority of the papers use the viscoelastic hemostatic assays as a predictor, i.e. a certain assay is elevated or lowered, which then is associated with a specific outcome. I suggest that, before the conclusion, a paragraph is added with a critical evaluation of the current literature, and where there are areas for future study. I suggest that, but I’m not a specialist in orthopedy, viscoelastic hemostatic assay-guided interventions may improve patient’s outcome. For example, in long bone fractures, hypercoagulability is associated with deep vein thrombosis. It would be interesting to see that if hypercoagulability is present and if heparin or coumadin is administered, will it reduce the risk for deep vein thrombosis? Are there other general interventions that could improve the clinical outcome of orthopedic patients with the aid of viscoelastic hemostatic assays?
The activators should be explained in a bit more detail in paragraph 1.3. I don’t think you should mention all assays (ROTEM has a specific assay for aspirin for example), but you should indicate which extrinsic or cascade-targeted activation reagents are being used. At east describe the activators for the tests that are being discussed in the paper. ROTEM uses tissue factor for EXTEM as does the rapidTEG, for example.
Overall, this appears to be a concise and helpful review of viscoelastic hemostatic assays applied in a specific medical area.
Specific comment:
Line 228: (79 patients): this should probably be “79 cardiac patients”? Please adjust if needed
